# Evaluating the Adversarial Robustness of a Foveated Texture Transform Module in a CNN

**Jonathan Gant, Andrzej Banburski, Arturo Deza**
Center for Brains, Minds & Machines
Massachusetts Institute of Technology
{jongant,kappa666,deza}@mit.edu

## Abstract

Biologically inspired mechanisms such as foveation and multiple fixation points have previously been shown to help alleviate adversarial examples (Reddy et al., 2020). By mimicking the effects of visual crowding present in human vision, foveated, texture-based computations may provide another route for increasing adversarial robustness. Previous statistical models of texture rendering (Portilla & Simoncelli, 2000; Gatys et al., 2015) paved the way for the development of a Foveated Texture Transform (FTT) module which utilizes localized texture synthesis in foveated receptive fields (Deza et al., 2019). The FTT module was added to a VGG-11 CNN architecture and ten randomly initialized networks were trained on 20 class subsets of the Places and EcoSet datasets for scene and object classification respectively. The trained networks were attacked using Projected Gradient Descent (PGD) and the adversarial accuracies were calculated at multiple epochs to evaluate changes in robustness as the networks trained. The results indicate that the FTT module significantly improved adversarial robustness for scene classification, especially when the validation loss was at a minimum. However, the FTT module did not provide a statistically significant increase in adversarial robustness for object classification. Furthermore, we do not find a trade-off between accuracy and robustness (Tsipras et al., 2018) for the FTT module suggesting a benefit of using foveated, texture-based distortions in the latent space during learning compared to non-perturbed latent space representations. Finally, we investigate the nature of latent space distortions with additional controls that probe other directions in the latent space that are not texture-based. A link to our code is available at https://github.com/JonGant/FoveatedTextureTransform.

## 1 Introduction

Investigating the representational advantage of visual crowding is a theme of ongoing work (Rosenholtz et al., 2019). While there are a handful of models that simulate crowding via texture-based summary synthesis models (Freeman & Simoncelli, 2011; Rosenholtz et al., 2012; Deza et al., 2019; Wallis et al., 2019), there is still no general consensus on whether this spatially-adaptive computation serves a representational goal—or if it is a limitation of the human visual system—though some efforts have been made to find an answer (Cheung et al., 2016; Deza & Konkle, 2020; Reddy et al., 2020; Han et al., 2020). Crowding is a phenomena that is mainly noticeable in the periphery (away from the fovea) of the visual field, where image features are pooled together giving rise to a locally "crowded" representation of a visual stimulus (Figure 1C).

In an effort to investigate the representational goal of crowding, we designed and tested a fully-differentiable, foveated visual system that performs localized, texture-based computations (Balas et al., 2009; Wallis et al., 2016, 2019; Vacher et al., 2020; Ziemba & Simoncelli, 2021)—the same

3rd Workshop on Shared Visual Representations in Human and Machine Intelligence (SVRHM 2021) of the Neural Information Processing Systems (NeurIPS) conference, Virtual.

that give rise to the effects of crowding—and test its general accuracy and adversarial robustness on a scene and object classification task. Our hypothesis is that foveated, texture-based systems will have a lower baseline accuracy than a non-foveated system but greater adversarial robustness, potentially inferring a representational goal of crowding. We will empirically show that crowding simulated through this texture-based perturbation—that is implemented with the aid of a stylization operator on Gaussian noise in the latent space (Huang & Belongie, 2017)—regularizes the learned representation of the image leading to an increase in adversarial robustness contingent on the dataset (scenes vs. objects).

## 2 Design of the Foveated Texture Transform (FTT)

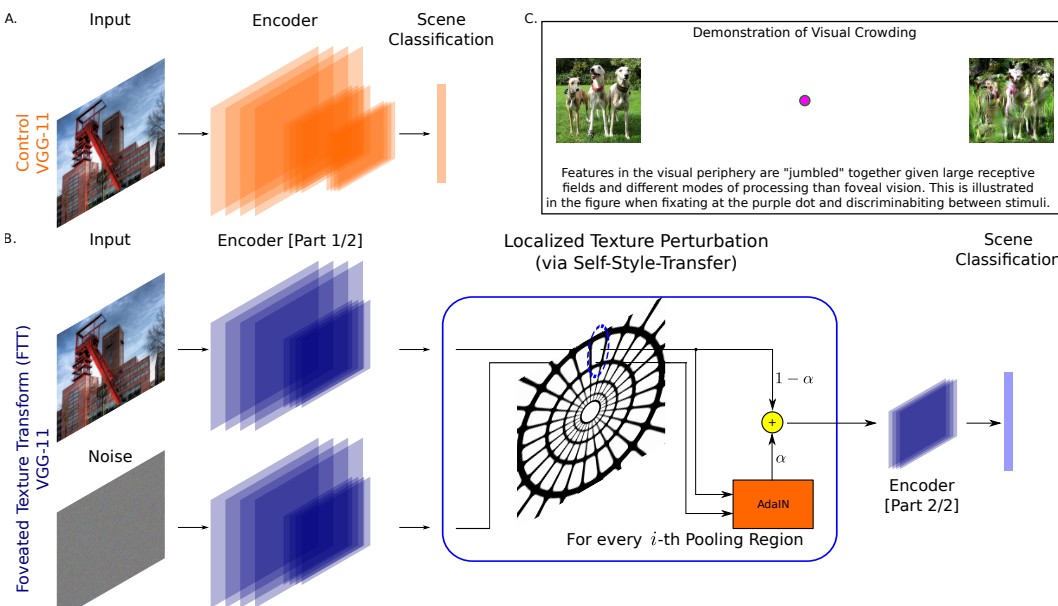

Figure 1: A schematic of the two main networks used in our experiments. We use a regular VGG-11 as our control system (A.) that performs an image classification task; and a foveated, texture-based VGG-11 where foveation happens in the latent space of the network (B.). (C.) Example of the visual crowding phenomena. When fixating on the middle magenta dot the two images in the periphery appear identical despite the distortions in the right image. The image on the right was generated through a foveated, texture-based synthesis model (Freeman & Simoncelli, 2011). Our transform approximates this type of texture-based computation in the latent space of a VGG-11.

Current models of texture-based foveation are two-stage such as those introduced in Deza & Konkle (2020). While these models succeed at probing the emergent representations of a particular type of computation such as adaptive Gaussian blurring that simulates a loss of visual acuity, or adaptive texture-based distortions that simulates crowding; a limitation—or advantage depending on the hypothesis being tested—is that two-stage models are not end-to-end differentiable by design. Within the context of evaluating adversarial robustness, this is a limitation and only few groups have succeeded in testing foveated models (Luo et al., 2016; Reddy et al., 2020; Jonnalagadda et al., 2021). These non-differentiable two-stage models are generally defined by a *fixed* first-stage transform $f(\circ)$, and a *learnable* second-stage transform $g(\circ)$ such that the general perceptual system $S$ that receives an image $I$ is defined as

$$S = g(f(I)), \tag{1}$$

where $f$ is an image-to-image transform as done in Deza & Konkle (2020) resembling the form $f = \mathcal{D}(\mathcal{E}_\Sigma(\circ))$ with the foveated encoder $\mathcal{E}_\Sigma$ and decoder ($\mathcal{D} \sim \mathcal{E}^{-1}$). Critically, in this paper we construct a fully differentiable model by removing $\mathcal{D}$ and making $S$ two-stage by partitioning the VGG-11 encoder into two parts ($f' : \mathcal{E}^{(1)}, g' : \mathcal{E}^{(2)}$). The foveation transform ($\circ_\Sigma$) was then embedded in the latent space between the two parts of the VGG-11 encoder at convolutional layer 4-1. The choice of applying the transformation after convolutional layer 4-1 was based on previous

work on texture synthesis (Huang & Belongie, 2017). Thus, our foveated network takes the following shape (similar to Eq. 1) $S = \mathcal{E}^{(2)}(\mathcal{E}_\Sigma^{(1)}(\circ))$. By de-coupling the foveated stylization function ($\mathcal{S}_\Sigma$) from the encoder we can now define the VGG-11 Foveated Texture Transform (FTT) as end-to-end differentiable where the encoders ($\mathcal{E}^{(1)}, \mathcal{E}^{(2)}$) are learnable via:

$$S = \mathcal{E}^{(2)}(\mathcal{S}_\Sigma(\mathcal{E}^{(1)}(I), \mathcal{E}^{(1)}(N))). \tag{2}$$

It is worth noting that the stylization function we use

$$\mathcal{S}(n, m) = \sigma(n)\left(\frac{m - \mu(m)}{\sigma(m)}\right) + \mu(n) \tag{3}$$

was introduced in Huang & Belongie (2017) and consists of aligning the channel-wise mean and standard deviation of the hidden feature activations in a deep neural network of a content image $m$ with a style image $n$. When the content image is noise, the stylization function can be used as a feed-forward texture synthesis model. Its foveated version ($\mathcal{S}_\Sigma$) was introduced in Deza et al. (2019) and consists of the same operation applied over log-polar receptive fields in the latent space. A diagram comparing our VGG-11 FTT model to the control VGG-11 can be seen in Figure 1.

## 3  Implications of FTT on Adversarial Robustness

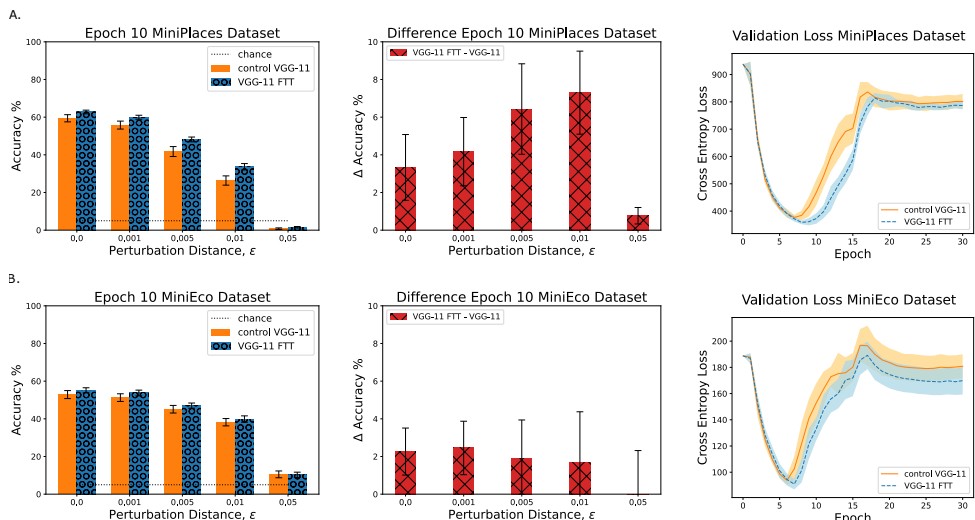

Figure 2: Our results suggest that VGG-11 FTT network shows greater robustness to $L_\infty$ PGD attacks compared to the control VGG-11 network – however this result is only noticeable for scene (A.) but not object (B.) classification. The observed increase in adversarial robustness for scene classification is highest around epoch 10 when the validation loss of the model is near a minimum (A.). Additional accuracy-robustness tradeoff curves can be found in Appendix B.

Both the control VGG-11 and VGG-11 FTT networks were trained for 30 epochs on a 20 class subset of the Places (Zhou et al., 2017) and EcoSet (Mehrer et al., 2021) datasets for scene and object classification respectively. Ten unique, random initializations of each network were trained to provide a large enough sample size for statistical analyses of the results. Appendix A contains more information on the training hyperparameters used.

After training, 5-step untargeted $L_\infty$ PGD attacks (Carlini & Wagner, 2017) were applied to the control VGG-11 and VGG-11 FTT networks on both datasets at epochs 10 and 20 across a range of $\epsilon \in \{0, 0.001, 0.005, 0.01, 0.05\}$ with a step size of $\frac{\epsilon}{3}$ using the Foolbox Python library (Rauber et al., 2017). As seen in Figure 2A, for scene classification the VGG-11 FTT network has a higher baseline accuracy and adversarial accuracies than the control VGG-11 network. The difference in the network performances is largest at $\epsilon = 0.01$ for the PGD attacks performed at epoch 10. Importantly, at epoch 10 the validation loss for both networks is near a minimum. At higher epochs, the differential advantage of the FTT module for scene classification decreases but is still apparent (Appendix B:

Figure 7). On the other hand, for object classification the FTT module does not provide an appreciable increase in adversarial accuracy across all epochs investigated (Figure 2B, Appendix B: Figure 8).

# 4 Contributions of FTT to Image Representation

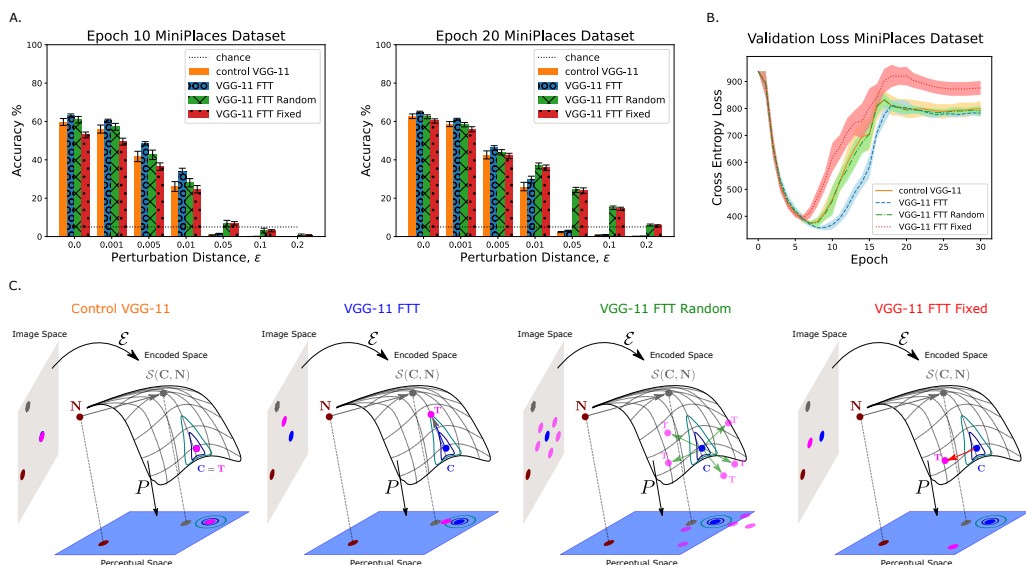

Figure 3: (A.) $L_\infty$ PGD attack results for VGG-11 FTT Random and Fixed networks. (B.) Cross-entropy validation loss values for all networks up to epoch 30. (C.) Visualization of the perturbation directions in the latent space for each network.

In addition to the FTT, another module with a similar transform to the FTT was developed to determine whether stochasticity or texture is the primary contributing factor to the increase in adversarial robustness (Dapello et al., 2020; Berardino et al., 2017; Dapello et al., 2021). In this transformation the difference of the stylized noise feature vector with the content feature vector is shuffled to maintain the magnitude of the transformation while making the perturbation direction in the latent space random. The ($i$-th) receptive field or pooling region in the FTT is defined by

$$T_i(I, N, \alpha_i) = (1 - \alpha_i)\mathcal{E}_i^{(1)}(I) + \alpha_i \mathcal{S}\left(\mathcal{E}_i^{(1)}(I), \mathcal{E}_i^{(1)}(N)\right). \tag{4}$$

Reformulating the above equation as

$$T_i(I, N, \alpha_i) = \underbrace{\mathcal{E}_i^{(1)}(I)}_{\text{Initial Encoding}} + \underbrace{\alpha_i\left(\mathcal{S}\left(\mathcal{E}_i^{(1)}(I), \mathcal{E}_i^{(1)}(N)\right) - \mathcal{E}_i^{(1)}(I)\right)}_{\text{Perturbation Vector}} \tag{5}$$

highlights the interpretation of the FTT as a perturbation to the encoded content in a direction defined by the difference of the stylized noise (texture-equivalent) and encoded content. The magnitude of this perturbation for each receptive field is controlled by $\alpha_i$. Let

$$D(I, N) = \mathcal{S}\left(\mathcal{E}_i^{(1)}(I), \mathcal{E}_i^{(1)}(N)\right) - \mathcal{E}_i^{(1)}(I) \tag{6}$$

and recognize that $D(I, N)$ is defined by a set of matrices whose entries can be enumerated by an index set $J = \{j \in \mathbb{N} | d_j \in D(I, N)\}$. Hence, a new direction with the same magnitude as $D(I, N)$ can be defined by permuting the index set and thus shuffling the elements of $D(I, N)$. The permutation of the index set can be randomly chosen each time the network performs a forward pass (Figure 3C: VGG-11 FTT Random) or remain fixed for a given network (Figure 3C: VGG-11 FTT Fixed). Importantly, since the VGG-11 FTT Random network uses a new permutation every forward pass, the same image can elicit different activations in the network. In contrast, the VGG-11 FTT Fixed network produces the same response every time for a given image. The VGG-11 FTT Fixed

condition was included as a control to determine how the extra stochasticity in the VGG-11 FTT Random network impacts accuracy and adversarial robustness.

Ten unique, random initializations of the VGG-11 FTT Random and Fixed networks were trained with the same hyperparameters used to train the control VGG-11 and VGG-11 FTT networks. After training, 5-step $L_\infty$ PGD attacks were performed on the networks and the adversarial accuracies are reported in Figure 3A. At epoch 20, the performance of the VGG-11 FTT Random network is comparable to the control VGG-11 at low perturbation distances and retains higher accuracies as the perturbation distance $\epsilon$ increases when compared to the control VGG-11 and VGG-11 FTT networks. However, at epoch 10 the VGG-11 FTT Random network closely matches the accuracy of the control VGG-11 network but has significantly lower adversarial robustness compared to the VGG-11 FTT network. At all perturbation distances, the VGG-11 FTT Fixed network has a marginally lower accuracy than the VGG-11 FTT Random network at epoch 20 and this difference is larger at epoch 10. Interestingly, the validation loss of all of the networks follows a similar behavior as seen in Figure 3B. Hence, the VGG-11 FTT Random network performs better when the validation loss is *not* at a minimum in contrast to the VGG-11 FTT network.

Additionally, representation similarity analysis was performed on the control VGG-11 and VGG-11 FTT networks at epoch 10. The feature vector at the end of convolutional layer 4-1 in the control VGG-11 and VGG-11 FTT networks were compared using the cosine similarity metric. The results indicate that between the same network the learned representations of each class had a similarity close to one when compared to itself on both datasets (Appendix C). Notably, the similarity across the diagonal is not exactly one since multiple images from the same class have different representations in the network. On the MiniPlaces dataset, the similarity between different classes between the same network is also quite low (Appendix C). Hence, the learned representations of each class in the MiniPlaces dataset in both networks are distinct from each other. Interestingly, this result does not hold for the VGG-11 FTT on the MiniEco dataset. The similarity of representations between different object classes in the VGG-11 FTT compared to itself was relatively close to one and greater than in the control VGG-11 (Appendix C) indicating a lack of distinction between the representations of object classes. When comparing between the control VGG-11 and the VGG-11 FTT it is clear that the learned representations are dissimilar on both datasets between all classes (Appendix C).

PGD Attack w/ ε=0.5

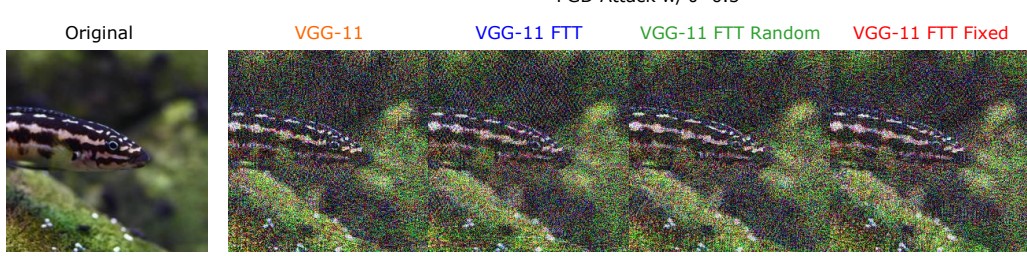

Figure 4: Visualization of an adversarial example for the VGG-11, VGG-11 FTT, VGG-11 FTT Random, VGG-11 FTT Fixed networks at epoch 10 for an exaggerated perturbation distance of $\epsilon = 0.5$ on an image of an aquarium from the MiniPlaces dataset.

Finally, we inspected if the direction of the perturbation in the latent space causes perceptual differences in the adversarial examples for each network. There are not any significant differences in terms of the visual structure of the PGD attacks across different networks (Figure 4). This is a surprising qualitative result, as we had expected that the adversarial noise of the other models that are not the control VGG-11 (VGG-11 FTT, VGG-11 FTT Random and VGG-11 FTT Fixed) to be either perceptually aligned (Santurkar et al., 2019) or spatially-adaptive (Reddy et al., 2020).

## 5   Discussion

The results indicate that the FTT significantly increased adversarial robustness for scene classification across all epochs investigated (Figure 2A, Appendix B: Table 1, Table 2). In particular, the FTT showed the largest advantage for scene classification when the validation loss was near a minimum

around epoch 10. Additionally, there is no trade-off between accuracy and robustness (Tsipras et al., 2018) when the FTT module is added to the VGG-11 network for scene classification. The lack of trade-off showcases a benefit of using foveated, texture-based distortions in the latent space instead of non-perturbed latent space representations during learning.

In contrast to the performance of the FTT on scene classification, the FTT did not provide a statistically significant increase in adversarial robustness for object classification (Figure 2B, Appendix B: Table 3, Table 4). The difference in performance between scene and object classification could be due to the lack of features in the periphery in object classification images. Additionally, scenes tend to be in natural settings and contain repetitive textures while objects are usually made by humans and lack repeating elements. The lack of repeating elements results in a poor representation when texture-based computations are performed as shown in Gatys et al. (2015). Previous psychophysical work has also shown that humans can recognize scenes accurately with texture-based representation (Renninger & Malik, 2004), whereas the same has not been shown for objects, suggesting that texture-based representations are able to encapsulate the information necessary for scene but not object classification.

Moreover, the VGG-11 FTT Random and Fixed networks have higher adversarial robustness at higher epochs and perturbation distances on the MiniPlaces dataset (Figure 3A, Appendix B: Table 2) when compared to the reduced adversarial robustness of the VGG-11 FTT network. The difference in adversarial robustness for the VGG-11 Random and Fixed network is negligible suggesting the permutation itself and not the randomness of the permutation is the primary contributing factor to the increase in adversarial robustness. Overall, the results suggests that different types of latent space perturbations (texture and non-texture based) may implicitly act as regularizers of learned visual representations.

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

## A  Training Dynamics and Convergence of Loss Function

The networks were trained using Stochastic Gradient Descent (SGD) with a weight decay of $5 \times 10^{-4}$ and momentum of 0.9. To ensure convergence of the loss function, a learning rate scheduler was used. The initial learning rate of $10^{-3}$ was set to $5 \times 10^{-4}$ at epoch 15 and $2.5 \times 10^{-4}$ at epoch 30. A batch size of 16 was used, the images were resized to 256x256 and color normalized. The number of training images were the same for both datasets to ensure consistency in the training dynamics.

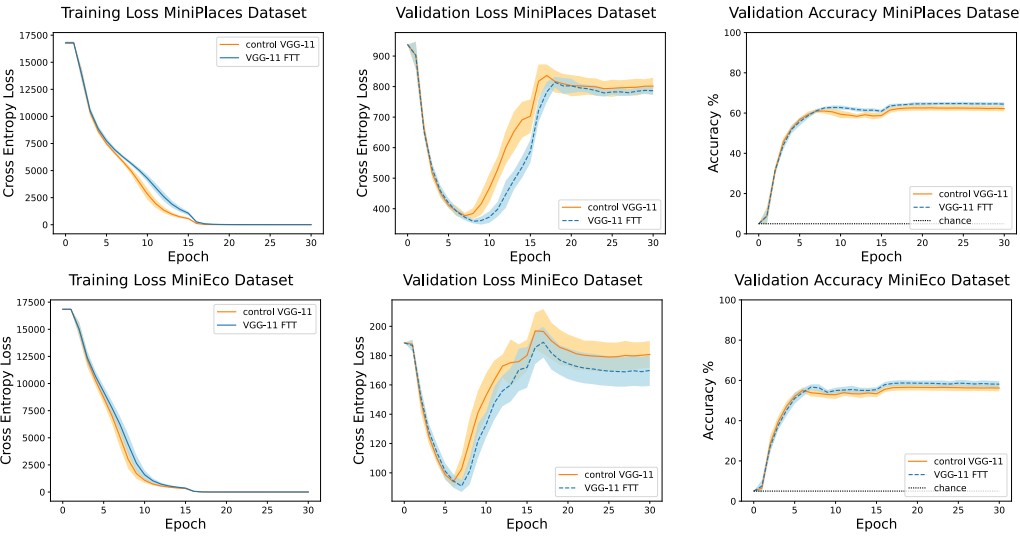

Figure 5: Cross entropy training/validation loss and accuracy as a function of epoch for both VGG-11 and VGG-11 FTT networks trained on the MiniPlaces (top) and MiniEco (bottom) datasets.

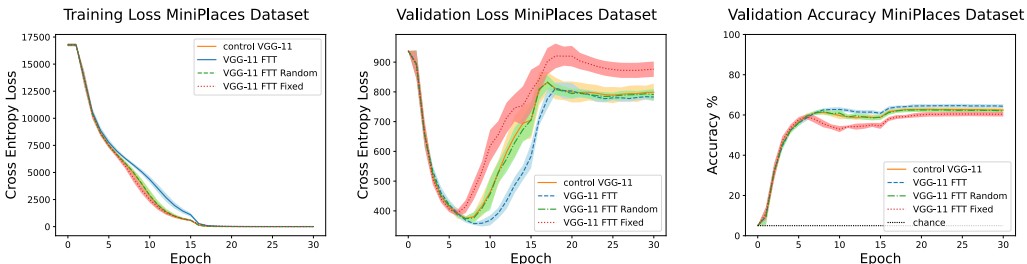

Figure 6: Cross entropy training/validation loss and accuracy as a function of epoch for control VGG-11, VGG-11 FTT, VGG-11 FTT Random, VGG-11 FTT Fixed networks trained on the MiniPlaces dataset.

## B  Adversarial Robustness as a Function of Learning Dynamics

| Epoch | $\epsilon = 0$ | $\epsilon = 0.001$ | $\epsilon = 0.005$ | $\epsilon = 0.01$ | $\epsilon = 0.05$ |
|---|---|---|---|---|---|
| 10 | $2.6 \times 10^{-5}$ | $5.6 \times 10^{-6}$ | $8.5 \times 10^{-8}$ | $5.9 \times 10^{-8}$ | $1.1 \times 10^{-10}$ |
| 20 | $4.2 \times 10^{-5}$ | $4.4 \times 10^{-6}$ | $2.2 \times 10^{-5}$ | $7.6 \times 10^{-5}$ | $9.1 \times 10^{-5}$ |
| 30 | $1.6 \times 10^{-5}$ | $4.0 \times 10^{-6}$ | $3.7 \times 10^{-6}$ | $4.9 \times 10^{-5}$ | $1.2 \times 10^{-5}$ |

Table 1: p-values from two-tailed t-test on the control VGG-11 and VGG-11 FTT networks accuracies at different perturbation distances and epochs on the MiniPlaces dataset.

| Epoch | VGG-11 | VGG-11 FTT | VGG-11 FTT Random | VGG-11 FTT Fixed |
|---|---|---|---|---|
| 10 | 0.9572 | 1.227 | 1.508 | 1.422 |
| 20 | 1.081 | 1.243 | 3.656 | 3.549 |
| 30 | 1.046 | 1.190 | 3.569 | 3.479 |

Table 2: Area under the curve estimates for accuracies at various perturbation distances and epochs on the MiniPlaces dataset for the control VGG-11, VGG-11 FTT, VGG-11 FTT Random, and VGG-11 FTT Fixed networks.

| Epoch | $\epsilon = 0$ | $\epsilon = 0.001$ | $\epsilon = 0.005$ | $\epsilon = 0.01$ | $\epsilon = 0.05$ |
|---|---|---|---|---|---|
| 10 | $1.7 \times 10^{-2}$ | $9.6 \times 10^{-3}$ | $2.7 \times 10^{-2}$ | $4.9 \times 10^{-2}$ | $7.5 \times 10^{-1}$ |
| 20 | $1.1 \times 10^{-2}$ | $1.1 \times 10^{-2}$ | $5.3 \times 10^{-3}$ | $5.9 \times 10^{-2}$ | $9.8 \times 10^{-1}$ |

Table 3: p-values from two-tailed t-test on the control VGG-11 and VGG-11 FTT networks accuracies at different perturbation distances and epochs on the MiniEco dataset.

| Epoch | VGG-11 | VGG-11 FTT |
|---|---|---|
| 10 | 1.422 | 1.474 |
| 20 | 1.644 | 1.695 |

Table 4: Area under the curve estimates for control VGG-11 and VGG-11 FTT accuracies at various perturbation distances and epochs for the MiniEco dataset.

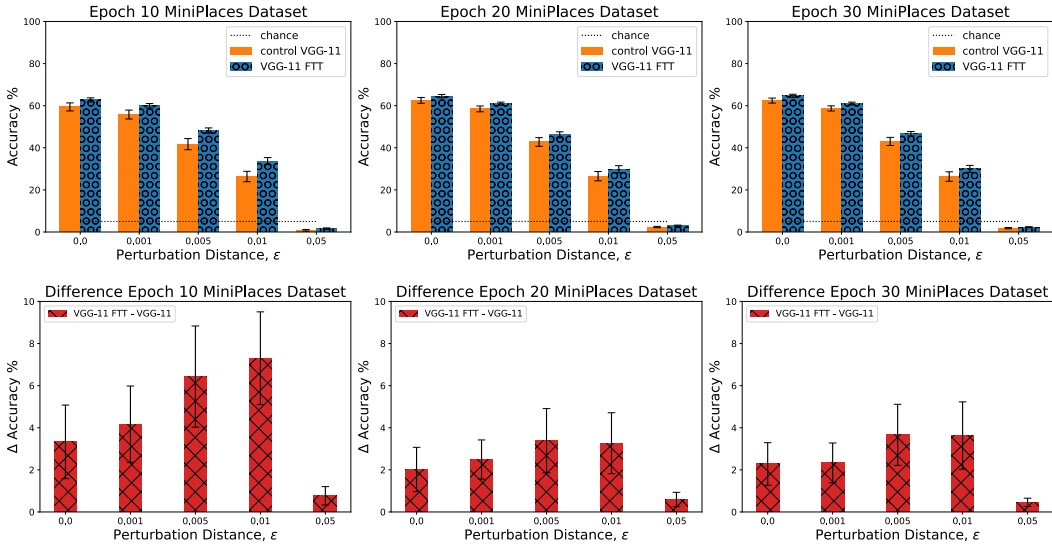

Figure 7: 5-step $L_\infty$ PGD attacks on control VGG-11 and VGG-11 FTT for the MiniPlaces dataset. The top row shows the performance of both models and bottom row shows the difference in accuracy across multiple epochs (10, 20, and 30).

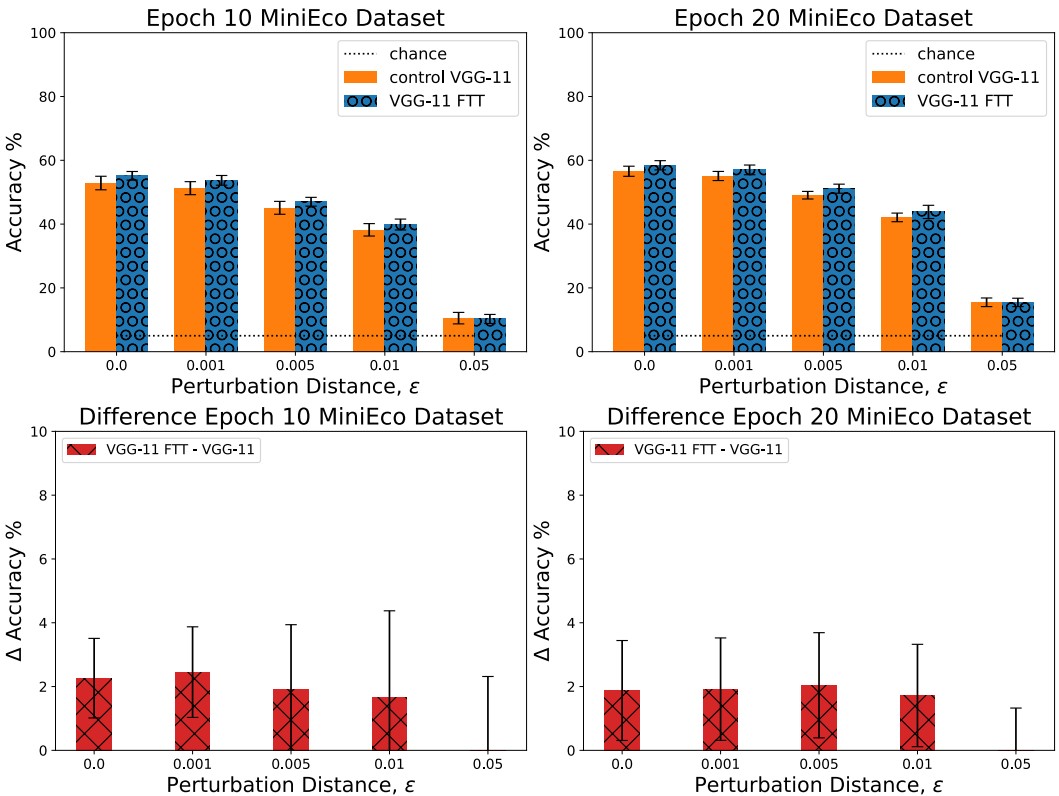

Figure 8: 5-step $L_\infty$ PGD attacks on control VGG-11 and VGG-11 FTT networks for the MiniEco dataset. Top row shows the performance of both models and bottom row shows the difference in accuracy.

## C Representation Similarity Analysis

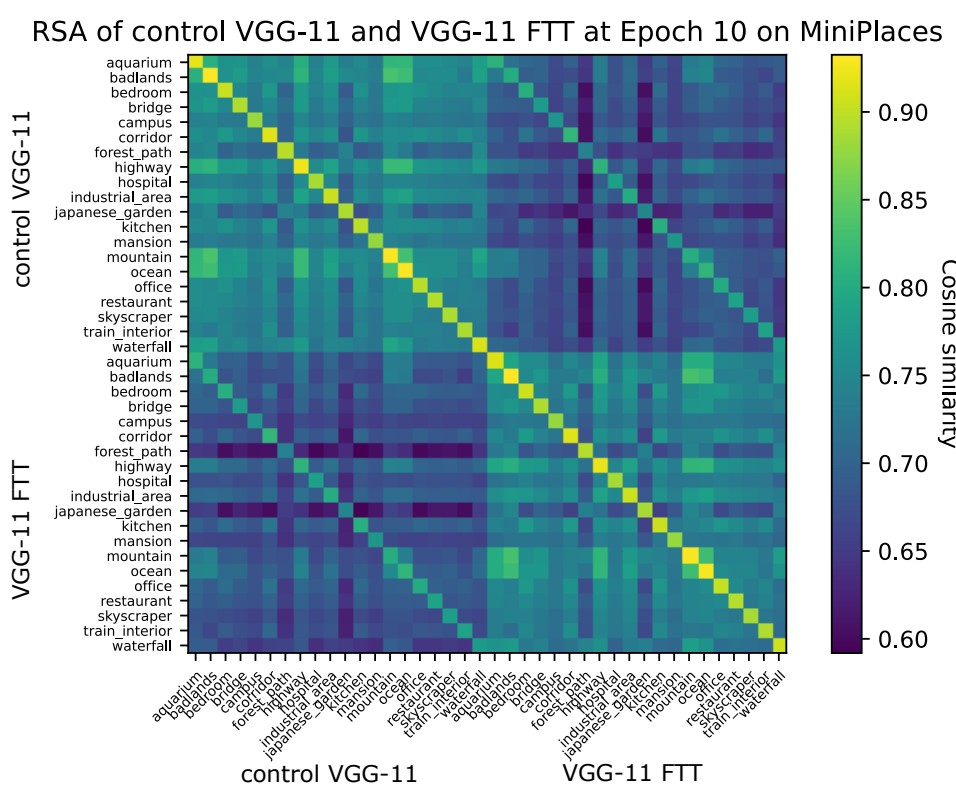

Figure 9: RSA results for the control VGG-11 and VGG-11 FTT on the MiniPlaces dataset for scene classification. Notably, the similarity across the diagonal is not exactly one since multiple images from the same class have different representations in the network and thus have a similarity that is less than one.

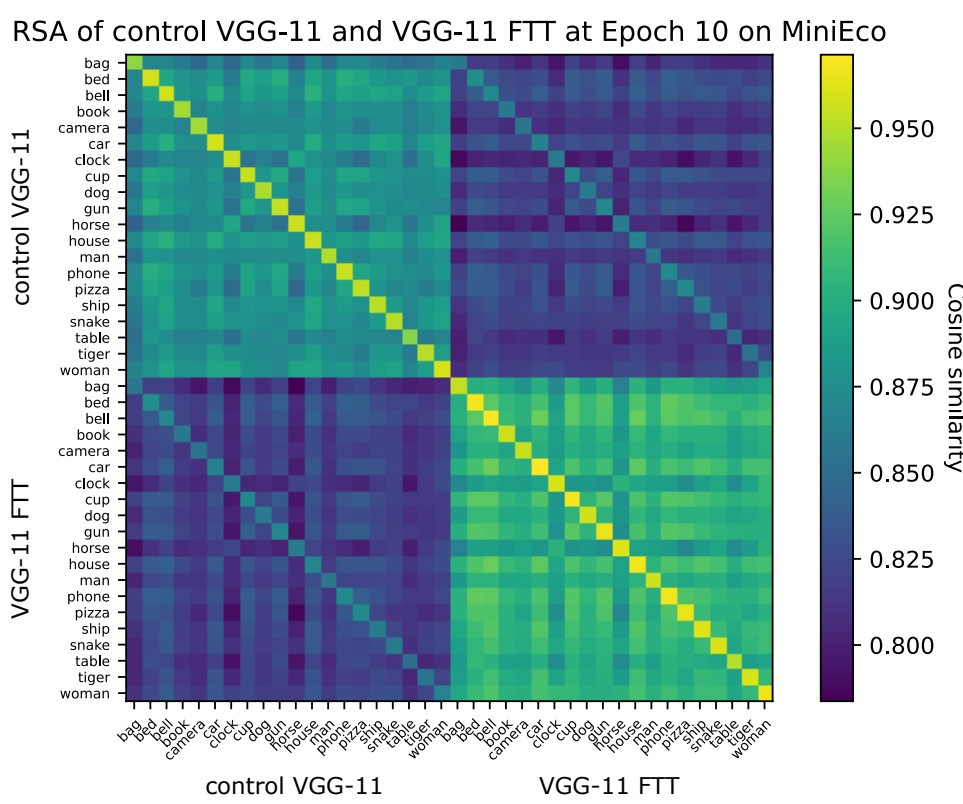

Figure 10: RSA results for the control VGG-11 and VGG-11 FTT on the MiniEco dataset for object classification. Notably, the similarity across the diagonal is not exactly one since multiple images from the same class have different representations in the network and thus have a similarity that is less than one.

