# OpenReview forum: "Evaluating the Adversarial Robustness of a Foveated Texture Transform Module in a CNN"
_NeurIPS.cc/2021/Workshop/SVRHM — SVRHM 2021 Poster_

### Official Review · Reviewer_6Mwc · 2021-10-29
**Interesting and solid study investigating the potential representational advantages of crowding**

**Rating:** 8
**Confidence:** 4

**Review:**

**Summary**
This study investigates the potential representational advantages of crowding - foveation or the lack of peripheral acuity. One hypothesis is that it could increase the adversarial robustness of models trained for object recognition and scene recognition. The authors thus test models with and without texture-based foveation (FTT) for both tasks. They find that the FTT models are indeed more robust for the scene recognition task, without any trade-of in terms of accuracy. They conclude that perturbations like the one studied here may thus implicitly act as regularizers of learned visual representations.
Overall, this paper is clearly written and convincing. The method used here is not novel (see Deza & Tonkle, 2021) but offers some nice additions and makes some interesting points. The scientific work is solid. I would have some minor suggestions, but I believe this work should be accepted.

**Strong points**
The method used here - the texture-based perturbations - is an interesting way to study the purpose of foveation. It is not novel to this work (see Deza & Tonkle, 2021) but offers 2 main additions: the FTT is now fully differentiable, and it is tested against adversarial attacks.
The paper is clearly written: no typos and well structured.

**Minor suggestions**
Here are minor suggestions, not in order of relevance:
 - please make the labels of the figures larger: there are really hard to read as they currently are
 - As far as I understood this work, it implements the model from (see Deza & Tonkle, 2021) with some modification - differentiability. I think this should be more clearly stated, potentially even in the introduction, at least in the "Design of the Foveated Texture Transform" section.
- I would have liked to see example of FTT images next to the original images (kind of like Fig. 1 in Deza & Tonkle, 2021). At least in the supplementary materials. The ex in Fig. 1 C is not foveated, and that's confusing. If possible, with examples for both dataset types (scenes and objects): This may give cues as to why there is a difference in the results between the object recognition and scene recognition tasks.
 - I don't really understand what's the point of the FTT-fixed condition. And extra, small explanation could be nice.

---

### Official Review · Reviewer_TyVb · 2021-11-01
**Interesting results that are well-aligned with workshop theme.**

**Rating:** 7
**Confidence:** 3

**Review:**

Summary:
This work uses a differentiable, Foveated Texture Transform module in a VGG-11 CNN and reports improved adversarial robustness in scene classification. Careful methodology, interesting results, well-aligned with workshop themes, so I recommend accept.

Comments:
1. The careful methodology (using 10 indep. runs, control experiment to evaluate the impact of stochasticity) is much appreciated.
2. The improvement in adversarial robustness is interesting, but it is not clear to me that this is *due* to visual crowding though the difference between scene and object classification are suggestive. It would be good to discuss at the workshop.

---

### Official Review · Reviewer_7L6H · 2021-11-01
**Interesting work with very promising results- few clarifications and issues remain that could help the presentation.**

**Rating:** 7
**Confidence:** 4

**Review:**

In this work the authors explore the adversarial robustness of modified deep networks that include a foveated texture transform (FTT) module in the latent space (within the network encoder). The authors train this model on both scene and object classification and then report results using a PGD adversarial attack to attack each of these trained models against their corresponding control models (deep network without the FTT module). They find that the FTT module significantly improves adversarial robustness in the scene classification problem, but not the object classification problem.

Pros:
-	The paper is well written and clear, although I would suggest making labels and axes on the plots larger since they were very hard to read.
-	I’m not totally familiar with the literature, but it sounds like the idea of placing the FTT model within the VGG network (applied to it’s latent space) and have the entire model be differentiable so that adversarial attacks can be measured etc. is in fact an original idea that has never been tested. I believe this is a valuable step in bringing these foveated texture models to the machine learning community and demonstrating their impact.
-	There is a definite clear benefit in terms of robustness on the scene classification task over the control VGG network.
-	The fact that there is no loss in the “clean” accuracy is very interesting and something that cannot be said for most models that defend against adversarial attacks.
-	I like the analysis of the permuted FTT perturbation vs the original and how the random/fixed permutation seems to help robustness at even higher epsilon  levels.

Areas for Improvement:
-	I think the model description could be more clear. It is unclear where in the VGG-11 network the FTT stage is added and what is the reasoning for the exact choice of where the pooling region is included? As I understand it, there is a single FTT module added in between two pieces of the VGG-11 encoder but I would think the results of these experiments would depend on exactly where the FTT module is placed and this should definitely be included in the final version if accepted.
-	The authors describe that the adversarial robustness is seen for scenes but not object recognition and while they provide a brief explanation of why this may occur, I would like to see more analysis on this issue. Is this an effect seen on just a subset of the images in the dataset that drives the overall accuracy or is this true for all of the “object” images? Can you show that the adversarial robustness for certain images (i.e. those that have more texture in the background) is increased more than other images?
-	I think the issue of overfitting and choosing models at different epochs is quite confusing for the paper. The authors claim that the original FTT models are robust at epoch 10 (where the validation loss is minimum), but the permuted FTT model shows improvement even beyond that indicating that overfitting is avoided. But I’m not sure I understand this statement because on the validation loss curves in Fig 3(b), all of the models clearly overfit in terms of the validation loss after epoch 10 (even if the robustness differs between those models). I think this makes the results much harder to interpret. I would suggest simply for all experiments using an early stopping rule where the best model is selected based on a convergence criteria of the validation loss (which will be around epoch 10). And comparing only those models. I’m not sure what the analysis of models at epochs beyond that point really provides.
-	I would like to ideally see this result on at least one other deep network architecture. We know that deep network architectures can change results quite a bit and to show the generality of this result, it would be nice to see that the FTT module can be inserted into at least one other architecture and also show this adversarial robustness.

---

### Decision · Program_Chairs · 2021-11-02

Accept (Poster)